**Data Availability Statement:** The HELIUS data are owned by the Amsterdam University Medical Centers, location AMC in Amsterdam, The

# Higher prevalence of depressed mood in immigrants' offspring reflects their social conditions in the host country: The HELIUS study

**Karien Stronks**[1]*, **Aydın Şekercan**[1], **Marieke Snijder**[1], **Anja Lok**[2], **Arnoud P. Verhoeff**[3,4], **Anton E. Kunst**[1], **Henrike Galenkamp**[1]

1 Department of Public Health, Amsterdam UMC, University of Amsterdam, and Amsterdam Public Health Research Institute, Amsterdam, Netherlands, 2 Department of Psychiatry, Amsterdam UMC location AMC, University of Amsterdam, Amsterdam, Netherlands, 3 Department of Epidemiology, Health Promotion and Care Innovation, Public Health Service Amsterdam, Amsterdam, Netherlands, 4 Department of Sociology, University of Amsterdam, Amsterdam, Netherlands

* k.stronks@amsterdamumc.nl

## Abstract

### Background

Immigrants from low- and middle-income countries who have settled in high-income countries show higher risks of depression in comparison with host populations. The risks are associated with adverse social conditions. Indecisive results have been reported on the depression risks of the *offspring* of immigrant populations.

### Objective

To assess the prevalence of depressed mood in immigrant offspring relative to the host population and to analyse whether that risk is explained by social conditions.

### Methods

Cross-sectional data from the Dutch HELIUS study were analysed, involving 19,904 men and women of Dutch, South-Asian Surinamese, African Surinamese, Turkish or Moroccan ethnic descent aged 18 to 70. Depressive symptomatology was assessed using the Patient Health Questionnaire-9 (PHQ-9). Indicators of social conditions were socioeconomic position (educational level, occupational level, employment status), perceived ethnic discrimination and sociocultural integration (ethnic identity, cultural orientation, social network). We used logistic regression to assess the risk of depressed mood (PHQ-9 sum score ≥10) in immigrants' offspring, as well as in first generation immigrants, relative to the risk in the host population. Social indicators were stepwise added to the model.

### Results

The prevalence of depressed mood was 13% to 20% among immigrant offspring, with the lowest level for those of African Surinamese descent; prevalence in the Dutch origin

Netherlands. Any researcher can request the data by submitting a proposal to the HELIUS Executive Board as outlined at http://www.heliusstudy.nl/en/researchers/collaboration, by email: heliuscoordinator@amsterdamumc.nl. The HELIUS Executive Board will check proposals for compatibility with the general objectives, ethical approvals and informed consent forms of the HELIUS study. There are no other restrictions to obtaining the data and all data requests will be processed in the same manner.

**Funding:** The HELIUS study is funded by the University Medical Centers Amsterdam, location AMC, and the Public Health Service of Amsterdam, as well as by the Dutch Heart Foundation, the Netherlands Organization for Health Research and Development (ZonMw), the European Union (FP-7), and the European Fund for the Integration of non-EU immigrants (EIF).

**Competing interests:** The authors have declared that no competing interests exist.

population was 7%. Relative risk of depressed mood, expressed as average marginal effects (AMEs), decreased substantially in all offspring groups after adjustment for socio-economic indicators and discrimination. E.g. the AME of Turkish vs. Dutch decreased from 0.11 (0.08–0.13) to 0.05 (0.03–0.08). Patterns resembled those in first generation immigrants.

## Conclusions

Results suggest that the observed higher prevalence of depressed mood in immigrants' offspring will decline to the level of the host population as the various populations grow closer in terms of socioeconomic position and as immigrant offspring cease to experience discrimination.

## Introduction

Many immigrant groups from low- and middle-income countries who have settled in Europe show greater risks of depression than the host populations of their adopted countries. One study based on the European Social Survey has found higher levels of depressive symptoms in immigrants in most countries in northern and western Europe [1]. When immigrants are differentiated into specific ethnic groups, the pattern appears to hold for most groups, though prevalence rates diverge [2,3]. This disadvantaged position of immigrants in Europe seems in contrast with the position of immigrants in the United States of America and Canada. In these countries, a 'healthy immigrant paradox' has been reported, indicating a better mental health for immigrants as compared to the host population [4].

Studies have found that higher risks of depression in immigrant populations are strongly associated with adverse conditions in the host countries. At least three types of conditions have been shown to play a role. First, a lower socioeconomic position, reflected in indicators such as occupation level, education level or unemployment, is associated with depression [4–6]. Studies in several European countries indicate that the lower average socioeconomic position of immigrants in high-income countries can account for part of their higher risk of depressive symptomatology [1,7–9]. Second, individually perceived discrimination, defined as unfair treatment on grounds of ethnic background, has been found associated with depression risk [4,5,10]. It has been shown to partially explain the higher risk of depressed mood in immigrants [1,3]. Third, immigrants may undergo stress from living in a culture that differs from their background culture, or from adjusting to a new culture, and that stress may subsequently heighten their risk of depression. Such cultural challenges include differing gender roles, poor language skills, loss of culture and values or cultural discordance between parental (traditional, familial) values and more Western values espoused by their offspring [9,11]. Some empirical studies have confirmed the salience of such cultural factors in the higher risk of depressed mood [12].

Studies on the risk of depression in the *offspring* of immigrants have shown mixed results [8, 13–18], with some studies reporting an elevated risk as compared with the host population, whereas others found risks similar to those in host populations and lower than those in people who have migrated themselves, to be referred to here as *immigrants*. The number of studies is limited, however, and they cover heterogeneous populations [17].

So far, we do not know whether social conditions can explain higher risks of depression in immigrant offspring in ways similar to those identified for their parents' generation. Against the background of the decisive role that host-country social conditions play in depression risk for those of the first generation, we hypothesise that a higher risk of depression in their offspring vis-à-vis the host population will also reflect differential social conditions faced by groups of immigrant offspring, including socioeconomic position, experienced discrimination and sociocultural integration. While previous studies have studied aspects of these social conditions [11,18–21], we are not aware of studies that simultaneously addressed these three groups of conditions in relation to the burden of depression in immigrant offspring.

The aims of this study are, first, to describe the prevalence of depressed mood in immigrant offspring in comparison with that in the host population. Second, we analyse whether observed differences in prevalence are accountable to adverse social conditions, and in particular to a low socioeconomic position, experienced discrimination or cultural distance to the country of residence. Finally, we will analyse whether any observed patterns in depressed mood amongst immigrant offspring, and in the role of social conditions, resemble the corresponding patterns in immigrants.

## Data and methods

### Study design

The Healthy Life in an Urban Setting (HELIUS) study is a multi-ethnic cohort study conducted in Amsterdam in the Netherlands. It has been described in detail elsewhere [22,23]. In brief, baseline data collection took place from 2011 to 2015 and included residents of Amsterdam of various ethnic origins in the ages of 18 to 70. Participants from six ethnic groups–of Dutch, African Surinamese, South-Asian Surinamese, Turkish, Moroccan and Ghanaian ethnic descent–were randomly selected from the municipal population register, stratified by ethnicity to ensure roughly equal numbers from each group. Sociohistorical information on the ethnic minority groups included in the study is provided elsewhere [23]. Data were collected by questionnaire and in a physical examination, in which biological samples were also obtained. Participants who were unable to complete questionnaires in Dutch were provided questionnaires in English or Turkish or received assistance from a trained, ethnically matched interviewer.

For the current study, cross-sectional baseline data were analysed. From the total sample completing the baseline questionnaire ($N = 23,942$), we first excluded participants not belonging to the six ethnic groups listed above ($n = 586$). We also excluded all respondents of Ghanaian origin ($n = 2,484$), as they included only a small number of immigrant offspring. From the remaining 20,872 participants, we excluded those with missing scores on the depressive symptomatology ($n = 183$), on the acculturation measures ($n = 455$), on any of the three measures of socioeconomic position ($n = 257$), on Dutch language proficiency ($n = 16$) or on the discrimination scale ($n = 57$). Our final study sample thus consisted of 19,904 participants: 4,591 of Dutch origin, 3,213 of South-Asian Surinamese, 4,212 of African Surinamese, 3,800 of Turkish and 4,088 of Moroccan ethnic origin.

The Medical Ethics Committee of the Amsterdam Academic Medical Centre (AMC) approved the study protocols. Written informed consent was obtained from all participants.

### Variables

**Ethnicity.**   Ethnicity was defined for each participant on the basis of their country of birth and that of their parents–currently the most widely accepted indicator of ethnicity in the Netherlands [24]. Specifically, participants were defined as *immigrant offspring* if they were born in

the Netherlands but both parents were born abroad; *immigrants* were born abroad of one or two foreign-born parents. Of residents of Surinamese background, approximately 80% are of either African or South-Asian descent; we classified Surinamese subgroups according to their self-reported ethnic descent. Participants representing the *host population* in the sample were born in the Netherlands of two parents born in the Netherlands.

**Depressed mood.** Depressive symptomatology was assessed using the Patient Health Questionnaire-9 (PHQ-9), which records depressive symptoms during the preceding two weeks. The instrument consists of 9 items with 4 response options for each item (0–3: never, several days, more than half the days, nearly every day), yielding a sum score range of 0 to 27, with higher scores indicating more depressive symptomatology [25]. If one item was missing (1.7%), the mean score of the other eight items was used to replace it; if more than one item was missing, the variable was treated as missing. Depressed mood was considered present if a respondent had a PHQ-9 sum score of 10 or above. That is a commonly used cut-off value, with a good sensitivity and specificity for predicting major depressive disorder [26]. The PHQ-9 has been shown to measure the same concepts across all six ethnic groups included in this study, with no systematic between-group differences in the identification of depressive symptoms [27].

**Social conditions in the host country.** *Socioeconomic position (SEP)* was assessed through educational level, occupational level and employment status. We included multiple indicators to cover as many socioeconomic influences as possible.

Participants were asked to report their most recent education and occupation. *Educational level* was the highest level attained in either the Netherlands or in the country of origin. It was distinguished into four categories: primary or less, lower secondary (general or vocational), upper secondary (general or vocational) and higher education. *Occupational level* was classified in five categories: elementary, lower, intermediate, higher or academic. Respondents' reported job titles and job descriptions, including any executive-level functions, were rated according to the Dutch Standard Occupational Classification (SBC-2010) [28], an extensive, systematic list of occupations in Dutch society. Cases for which the question on paid job were not relevant because they never had a paid job, were retained in the analysis as a separate group ("not applicable"), given their large numbers (from 6% in the Dutch subgroup to 27% in the Moroccan subgroup). *Employment status* was categorised as employed, unemployed (seeking work or on welfare benefit), not in labour force (student, retired, full-time homemaker) and incapacitated.

**Perceived ethnic discrimination.** Perceived ethnic discrimination (PED) was measured using the Everyday Discrimination Scale (EDS). It captures the frequency of discriminatory experiences in daily life, recording nine items (e.g. 'treated with less respect than others') on a 5-point Likert scale (never to very often) [29]. We adapted the EDS by specifically asking the participants how often they had experienced discrimination *because of their background*; in Dutch this phrase would be interpreted as ethnic background, especially in the context of this study. If one EDS item was missing, the mean score of the other items was used to replace it; if two or more items were missing, the variable was treated as missing. The prevalence of perceived ethnic discrimination was determined on the basis of respondents scoring 4 (often) or 5 (very often) on at least one item.

**Sociocultural conditions.** We based the assessments of sociocultural conditions upon several indicators that have been considered markers of the immigrants' position in the process of acculturation following migration, covering several domains of life: ethnic identity, cultural orientation and the ethnic composition of one's social network [30,11,31]. The measurement of these factors was based on the two-dimensional conceptualisation of acculturation by John W. Berry [30]. We assessed whether respondents were adapted to the host

society as well as whether they adhered to the culture of the original country. Four possible acculturation strategies derive those two dimensions: *integration* (high orientation to both host-country and original culture), *assimilation* (high orientation to host culture, low orientation to original culture), *separation* (high orientation to original culture, low orientation to host culture) and *marginalisation* (low orientation towards original as well as host culture).

Respondents were classified into one of the four acculturation strategies on the basis of their responses to statements relating to ethnic identity, cultural orientation and social network composition, each with five response categories ranging from 'totally disagree' to 'totally agree'. *Ethnic identity* was conceptualised as the sense of belonging to a particular ethnic group that shares cultural values and beliefs [32]. It was assessed with two items gauging the sense of belonging to the Dutch community and to the Turkish, Moroccan or Surinamese community (depending on background) [33]. *Cultural orientation* was measured using the 20 items of the Psychological Acculturation Scale (PAS; e.g. 'I have a lot in common with Dutch/Surinamese/Turkish/Moroccan people', 'I feel proud to be part of Dutch/Surinamese/Turkish/Moroccan culture') [33]. *Ethnic social network composition* was conceptualised in terms of the ethnic origin of friends, and of the people with whom one spends free time. It was measured by four items regarding the proportions of co-ethnics and ethnic Dutch people in one's social network and the proportion of free time spent with co-ethnic and ethnic Dutch people.

## Statistical analyses

Baseline sociodemographic characteristics and social conditions are reported for respondents from the host population, for immigrants and for immigrants' offspring, differentiated by ethnicity. Our first aim was to assess the prevalence of depressed mood in immigrant offspring relative to that in the Dutch host population. We used logistic regression models, controlling for age and gender, and obtained average marginal effects (AME) for each ethnic minority group [34]. AMEs were estimated using the margins post estimation command in STATA (College Station, TX: StataCorp LP). Effects of all predictors in the model are calculated for each observation in the dataset, and then averaged. The resulting average marginal effect can be interpreted as the average increase in the probability of depressed mood over all values of the covariates [35]. As an example, an average marginal effect of .20 indicates a 20% higher probability of depressed mood as compared to the reference group. We prefer AMEs above ORs, given that our aim is to compare the contribution of social determinants to the probability of depressed mood across groups, in particular generations as well as ethnic groups. When making comparisons across groups, ORs can be misleading as an indicator of the size of the contribution of each predictor as a large change in odds ratios does not always correspond with a large change in predicted probability. In contrast, AMEs do allow for a fair comparison of effect sizes across groups [34]. Finally, since the age ranges among immigrants and immigrants' offspring were very dissimilar, we did not directly compare the prevalence of depressed mood. Any differences between groups should be viewed with that in mind.

We also assessed the associations among immigrant offspring between indicators for social conditions (socioeconomic position, perceived discrimination, sociocultural conditions) and depressed mood, controlling for age and gender. Our second research aim was to analyse whether the distributions of social conditions accounted for the risks of depressed mood in immigrant offspring. To that end, we used logistic regression models to compare the risks of depression in the immigrant populations (both first generation and offspring) and the host population, adding the indicators for social conditions to the model both one by one, as well as a set (socioeconomic position, perceived discrimination, sociocultural conditions respectively). Because the host population had no values on the sociocultural indicators, we also

created models in each immigrant and offspring group based on what we considered the most favourable sociocultural profiles–integrated ethnic identity, integrated cultural orientation and integrated social network. We used decreasing AME as an indicator of the relative importance of a specific social condition for the higher depression risk.

To evaluate whether the patterns observed in immigrants' offspring resembled those in immigrants (our third research question), similar statistical analyses as described were performed on the immigrants in the sample.

Analyses were conducted using IBM SPSS Statistics for Windows (Armonk, NY: IBM Corporation) and STATA Statistical Software, Release 15 (College Station, TX: StataCorp LP). The level of statistical significance was set at $p < .05$.

## Results

The educational and occupational levels of the immigrants' offspring were lower than those of the Dutch host population (Table 1), but higher than that of the immigrants. The sociocultural profiles of the offspring in both Surinamese groups were similar to that of the co-ethnic immigrants on all three indicators. About 80% were classified as integrated on the ethnic identity and cultural orientation indicators, but much lower (36%–41%) on the social network indicator. In the Turkish and Moroccan groups, the offspring profiles differed from those of the immigrants themselves, with higher proportions of the offspring classified as integrated or assimilated. As expected, the percentages of people experiencing difficulties with the Dutch language were far lower for offspring than for immigrants in all ethnic groups. The immigrants' proficiency in Dutch varied widely between ethnic groups, with those from Suriname (a former Dutch colony) scoring high and those from Turkey and Morocco scoring low. Around one quarter (Turkish) to one third (other groups) of ethnic minority respondents reported having experienced discrimination, with similar figures for offspring and for immigrants.

The prevalence of depressed mood in immigrants' offspring was 13% to 20%, as compared with 7% in those of Dutch origin. It was lowest for African Surinamese and highest for Turkish offspring.

The outcomes of the logistic regression model, as expressed in average marginal effects (AMEs), confirmed the elevated risk of depressed mood in immigrant offspring (Table 2, first column). The AME for Turkish offspring, for example, was 0.11 vis-à-vis the Dutch population, indicating an 11% higher probability of depressed mood. Prevalence of depressed mood amongst the immigrant generations (second column) was 10% to 25%, slightly exceeding that of the offspring in three out of four ethnic groups. However, in direct comparisons between immigrant offspring and the first generation within a specific ethnic minority group (Table 2, final column), all confidence intervals except in the Moroccan groups included the possibility of equal depression risk in the offspring and the immigrant generations.

Table 3 shows the associations between all indicators for social conditions and depressed mood for the offspring of immigrants, controlling for age and gender, and stratified by ethnic group. Those in the lowest educational and occupational levels showed higher risks of depressed mood. In most ethnic groups, however, the associations between these two socio-economic indicators and depressed mood were not linear. The prevalence of depressed mood was clearly higher for people without paid jobs, in particular those who were incapacitated or outside the labour market. Immigrant offspring who had experienced ethnic discrimination showed 5% to 18% higher risks of depressed mood. With regard to the sociocultural factors, the social network variable, and to a lesser extent the cultural orientation variable, indicated higher risks of depressed mood for offspring classified as marginalised, and in some subgroups

**Table 1. Characteristics of immigrant offspring, immigrants and Dutch origin respondents in the study sample, *N* = 19904.**

| | South-Asian Surinamese | | African Surinamese | | Turkish | | Moroccan | | Dutch |
|---|---|---|---|---|---|---|---|---|---|
| | Offspring | Immigrants | Offspring | Immigrants | Offspring | Immigrants | Offspring | Immigrants | |
| | *n* = 802 | *n* = 2411 | *n* = 733 | *n* = 3479 | *n* = 1220 | *n* = 2580 | *n* = 1396 | *n* = 2692 | *n* = 4591 |
| **Gender (% female)** | 49.9 | 54.3 | 58.3 | 59.6 | 54.0 | 55.2 | 66.9 | 59.3 | 53.9 |
| **Age (years)** | 28.2 ± 6.6 | 50.4 ± 10.3 | 31.9 ± 9.2 | 50.7 ± 10.8 | 27.2 ± 6.5 | 45.6 ± 10.0 | 27.4 ± 6.2 | 46.0 ± 11.0 | 46.1 ± 14.0 |
| **Socioeconomic position** | | | | | | | | | |
| *Educational level* | | | | | | | | | |
| Higher education | 35.5 | 18.8 | 30.2 | 20.9 | 25.5 | 9.8 | 30.1 | 11.2 | 60.5 |
| Upper secondary | 45.3 | 25.7 | 47.9 | 33.8 | 44.2 | 22.6 | 49.9 | 26.7 | 22.0 |
| Lower secondary | 15.7 | 38.9 | 19.5 | 39.4 | 23.9 | 25.8 | 16.8 | 18.7 | 14.2 |
| Primary or less | 3.5 | 16.6 | 2.5 | 5.9 | 6.4 | 41.9 | 3.3 | 43.4 | 3.3 |
| *Occupational level* | | | | | | | | | |
| Academic | 6.7 | 3.8 | 2.0 | 2.7 | 4.5 | 2.3 | 4.1 | 1.3 | 19.8 |
| Higher | 22.1 | 13.9 | 21.1 | 16.7 | 13.1 | 5.9 | 18.2 | 7.8 | 36.4 |
| Intermediate | 25.9 | 28.7 | 32.3 | 32.1 | 24.8 | 16.5 | 29.4 | 17.2 | 22.1 |
| Lower | 27.2 | 32.5 | 26.1 | 34.4 | 33.8 | 30.7 | 26.6 | 24.9 | 14.3 |
| Elementary | 3.1 | 11.2 | 4.5 | 7.2 | 4.3 | 19.7 | 3.6 | 17.3 | 1.6 |
| Not applicable | 15.0 | 9.8 | 13.9 | 6.9 | 19.5 | 25.0 | 18.1 | 31.5 | 5.7 |
| *Employment status* | | | | | | | | | |
| Paid employment | 66.0 | 59.5 | 64.8 | 62.1 | 62.4 | 48.5 | 61.7 | 42.2 | 73.9 |
| Unemployed | 19.0 | 12.9 | 16.6 | 10.9 | 23.9 | 23.4 | 22.5 | 30.1 | 17.4 |
| Not in labour market | 12.0 | 16.4 | 14.5 | 16.9 | 10.0 | 16.6 | 12.5 | 17.3 | 5.6 |
| Incapacitated | 3.1 | 11.2 | 4.1 | 10.1 | 3.8 | 11.6 | 3.2 | 10.4 | 3.1 |
| **Discrimination** | | | | | | | | | |
| *Any discrimination* | 32.0 | 30.9 | 34.1 | 34.5 | 25.7 | 28.3 | 36.0 | 31.4 | 2.1 |
| **Sociocultural factors** | | | | | | | | | |
| *Ethnic identity* | | | | | | | | | |
| Integrated | 76.9 | 81.8 | 85.0 | 82.4 | 73.7 | 51.3 | 88.0 | 71.2 | – |
| Assimilated | 15.1 | 6.4 | 4.1 | 2.3 | 2.6 | 2.6 | 1.3 | 1.7 | – |
| Separated | 6.0 | 10.2 | 9.1 | 13.8 | 21.4 | 42.9 | 9.1 | 25.3 | – |
| Marginalised | 2.0 | 1.7 | 1.8 | 1.5 | 2.3 | 3.1 | 1.6 | 1.7 | – |
| *Cultural orientation* | | | | | | | | | |
| Integrated | 79.4 | 80.4 | 80.5 | 84.3 | 76.2 | 63.4 | 79.9 | 76.2 | – |
| Assimilated | 6.4 | 6.1 | 6.3 | 3.0 | 4.3 | 3.1 | 5.6 | 4.0 | – |
| Separated | 11.6 | 11.5 | 11.2 | 11.2 | 18.2 | 31.7 | 13.1 | 18.1 | – |
| Marginalised | 2.6 | 2.0 | 2.0 | 1.5 | 1.3 | 1.7 | 1.4 | 1.7 | – |
| *Social network* | | | | | | | | | |
| Integrated | 35.8 | 34.1 | 41.3 | 39.2 | 37.5 | 24.7 | 29.5 | 22.5 | – |
| Assimilated | 14.6 | 9.5 | 9.7 | 7.4 | 7.6 | 4.3 | 6.4 | 5.0 | – |
| Separated | 34.9 | 31.1 | 37.2 | 37.5 | 46.6 | 57.5 | 47.3 | 44.2 | – |
| Marginalised | 14.7 | 25.2 | 11.7 | 16.0 | 8.2 | 13.5 | 16.7 | 28.3 | – |
| *Difficulty with Dutch language (%)* | 10.1 | 25.5 | 7.8 | 13.0 | 21.6 | 74.4 | 9.3 | 61.3 | – |
| **Depressed mood** | 16.2 | 19.2 | 12.6 | 9.8 | 19.5 | 24.5 | 17.5 | 22.0 | 7.2 |

also for those classified as separated. In some groups, those classified as assimilated also showed higher depression prevalences than those in the integrated group. Across all social conditions, overall, the immigrant generation showed risk patterns similar to those of the offspring, with the risk being either higher or lower for specific indicators (S1 Table).

Table 2. Probability of depressed mood in immigrants' offspring and immigrants, relative to Dutch origin respondents, and relative to each other.

| | Average marginal effects (AMEs)* (with 95% confidence intervals) obtained from logistic regression models, with depressed mood as the outcome, controlling for age and gender | | |
|---|---|---|---|
| | *Immigrants' offspring relative to ethnic Dutch* | *Immigrants relative to ethnic Dutch* | *Immigrants' offspring relative to immigrants* |
| Dutch | 0.00 | 0.00 | – |
| South-Asian Surinamese | 0.09 (0.06, 0.11) | 0.12 (0.10, 0.14) | −0.04 (−0.09, 0.00) |
| African Surinamese | 0.05 (0.02, 0.07) | 0.03 (0.01, 0.04) | 0.01 (−0.02, 0.03) |
| Turkish | 0.11 (0.09, 0.14) | 0.17 (0.15, 0.19) | 0.00 (−0.04, 0.04) |
| Moroccan | 0.08 (0.06, 0.11) | 0.15 (0.13, 0.16) | −0.04 (−0.08, −0.00) |

*Average marginal effects (AME) can be interpreted as the higher/lower probability of having depressed mood as compared to the reference group

Finally, we analysed whether the distributions of social conditions accounted for the risks of depressed mood in immigrant offspring. To that end, we run regression models with depressed mood as the outcome, and immigrant offspring or immigrant status vis-à-vis the host population as the main predictor, controlling for age and gender. We then added the indicators for social conditions to the model, firstly, one by one, and secondly, as a set (for socioeconomic, ethnic discrimination and sociocultural factors respectively). We used decreasing AME as an indicator of the relative importance of a specific social condition for the higher depression risk. The outcome of these analyses are shown in S2 Table. Fig 1 depicts the results of the regression models for the combined set of indicators for socioeconomic position, ethnic discrimination and sociocultural factors respectively.

The higher risk of depressed mood in immigrant offspring vis-à-vis the Dutch host population is explained largely by adverse socioeconomic positions (Model 2 versus Model 1). In the Turkish group, for example, the AME decreased after control for socioeconomic indicators from 0.11 (95% CI 0.08–0.13) to 0.05 (95% CI 0.03–0.08). The contributions of each socioeconomic variable individually (education level, occupation level and employment status) were largely similar (S2 Table). After adding discrimination to the model, we no longer observed a significantly heightened risk of depressed mood in any group of immigrant offspring (Model 3). In immigrants, we found the contributions of social conditions to be largely similar to those observed in the offspring. However, in three out of four ethnic groups, the risk of depressed mood in immigrants remained elevated after control for differences in socioeconomic position and discrimination. Moroccan immigrants, for instance, still had an AME of 0.05 (95% CI 0.03–0.07). A small part of their remaining risk was accounted for by sociocultural variables, especially the level of social network integration (see S2 Table, Appendix, for results for subgroups based on the three sociocultural variables).

## Discussion

Our findings show that the prevalence of depressed mood was higher in immigrants' offspring (13% to 20%) as compared with people of Dutch origin (7%). The lowest levels of depressed mood were seen in offspring of African Surinamese descent and the highest in those of Turkish descent. Prevalence rates in immigrants' offspring were comparable to those in the immigrant generation in the corresponding ethnic group. The higher risk of depressed mood in immigrants' offspring relative to that in the host population was explained in particular by their adverse socioeconomic position, and to a lesser extent by perceived discrimination. The corresponding patterns in the first generation were largely comparable to those in the offspring,

**Table 3. Associations between indicators for social conditions and depressed mood in immigrants' offspring, by ethnic group.**

| | Average marginal effects[*] (with 95% confidence intervals) as obtained from logistic regression models with depressed mood as the outcome, and social conditions as main predictors, controlling for age and gender | | | |
|---|---|---|---|---|
| | **Ethnic group** | | | |
| | **South-Asian Surinamese** | **African Surinamese** | **Turkish** | **Moroccan** |
| **Socioeconomic conditions** | | | | |
| *Educational level* | | | | |
| Higher | 0.00 | 0.00 | 0.00 | 0.00 |
| Upper secondary | 0.03 (−0.03, 0.08) | 0.04 (−0.01, 0.09) | 0.06 (0.01, 0.11) | 0.06 (0.02, 0.11) |
| Lower secondary | 0.04 (−0.04, 0.12) | 0.09 (0.01, 0.16) | 0.07 (0.01, 0.13) | 0.13 (0.07, 0.20) |
| Primary or less | 0.05 (−0.10, 0.21) | 0.09 (−0.09, 0.27) | 0.24 (0.13, 0.35) | 0.18 (0.04, 0.32) |
| *Occupational level* | | | | |
| Academic | 0.00 | 0.00 | 0.00 | 0.00 |
| Higher | 0.04 (−0.05, 0.14) | −0.05 (−0.21, 0.11) | 0.10 (0.02, 0.19) | 0.04 (−0.04, 0.11) |
| Intermediate | 0.06 (−0.04, 0.16) | −0.00 (−0.16, 0.16) | 0.14 (0.06, 0.22) | 0.11 (0.04, 0.18) |
| Lower | 0.06 (−0.04, 0.16) | 0.03 (−0.14, 0.19) | 0.15 (0.08, 0.23) | 0.13 (0.05, 0.20) |
| Elementary | −0.07 (−0.20, 0.05) | 0.19 (−0.04, 0.42) | 0.14 (0.02, 0.27) | 0.13 (−0.00, 0.25) |
| Not applicable | 0.02 (−0.09, 0.13) | 0.01 (−0.16, 0.18) | 0.11 (0.03, 0.19) | 0.18 (0.10, 0.27) |
| *Employment status* | | | | |
| Employed | 0.00 | 0.00 | 0.00 | 0.00 |
| Unemployed | 0.01 (−0.06, 0.07) | 0.02 (−0.04, 0.08) | 0.02 (−0.04, 0.07) | 0.06 (0.01, 0.11) |
| Not in labour market | 0.11 (0.02, 0.20) | 0.13 (0.04, 0.21) | 0.16 (0.07, 0.25) | 0.15 (0.08, 0.22) |
| Incapacitated | 0.29 (0.10, 0.48) | 0.29 (0.12, 0.47) | 0.36 (0.22, 0.51) | 0.37 (0.22, 0.52) |
| **Discrimination** | | | | |
| No discrimination | 0.00 | 0.00 | 0.00 | 0.00 |
| Any discrimination | 0.05 (−0.01, 0.11) | 0.08 (0.02, 0.13) | 0.18 (0.13, 0.24) | 0.10 (0.06, 0.15) |
| **Sociocultural factors (ref: Integrated)** | | | | |
| *Ethnic identity* | | | | |
| Integrated | 0.00 | 0.00 | 0.00 | 0.00 |
| Assimilated | −0.01 (−0.08, 0.06) | −0.06 (−0.16, 0.04) | −0.05 (−0.17, 0.07) | 0.27 (0.04, 0.50) |
| Separated | 0.01 (−0.10, 0.13) | −0.02 (−0.10, 0.05) | −0.02 (−0.07, 0.03) | −0.05 (−0.11, 0.01) |
| Marginalised | −0.05 (−0.20, 0.10) | −0.06 (−0.20, 0.08) | 0.11 (−0.07, 0.29) | 0.07 (−0.12, 0.25) |
| *Cultural orientation* | | | | |
| Integrated | 0.00 | 0.00 | 0.00 | 0.00 |
| Assimilated | 0.08 (−0.03, 0.20) | 0.07 (−0.04, 0.19) | 0.15 (0.02, 0.28) | 0.09 (−0.01, 0.18) |
| Separated | 0.06 (−0.03, 0.14) | 0.04 (−0.03, 0.12) | 0.06 (0.00, 0.13) | 0.05 (−0.02, 0.11) |
| Marginalised | 0.15 (−0.05, 0.34) | 0.45 (0.20, 0.70) | 0.15 (−0.08, 0.38) | 0.13 (−0.06, 0.34) |
| *Social network* | | | | |
| Integrated | 0.00 | 0.00 | 0.00 | 0.00 |
| Assimilated | −0.01 (−0.08, 0.06) | −0.02 (−0.10, 0.07) | 0.08 (−0.01, 0.17) | 0.12 (0.02, 0.21) |
| Separated | 0.03 (−0.03, 0.09) | −0.01 (−0.06, 0.04) | 0.04 (−0.00, 0.09) | 0.02 (−0.02, 0.06) |
| Marginalised | 0.13 (0.05, 0.22) | 0.06 (−0.03, 0.15) | 0.12 (0.03, 0.22) | 0.10 (0.04, 0.17) |

[*]Average marginal effects can be interpreted as the higher/lower probability of having depressed mood as compared to the reference group

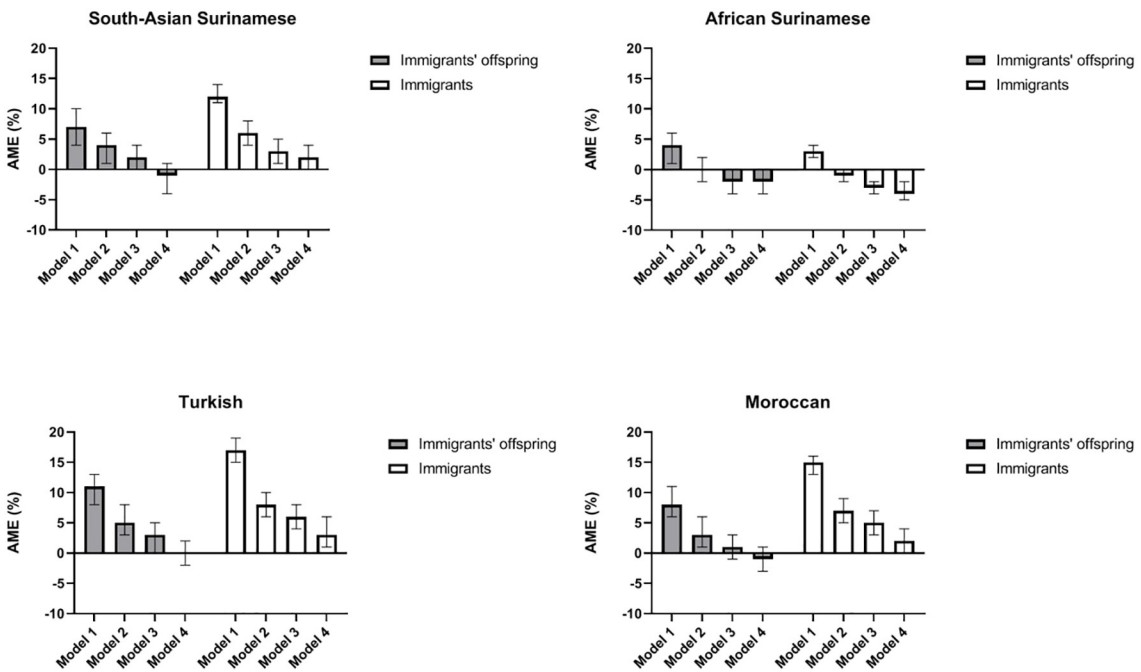

**Fig 1. Average Marginal Effects (AMEs) for depressed mood in immigrants' offspring and in immigrants, relative to those of Dutch origin, with stepwise adjustments for social conditions.** Model 1: age, gender; Model 2: age, gender, SEP; Model 3: age, gender, SEP, discrimination; Model 4: age, gender, SEP, discrimination, in Social network integrated.

although the immigrants' level of sociocultural integration also explained a small part of their observed higher risk of depressed mood.

## Limitations of the study design

As this study used cross-sectional data, we cannot exclude the possibility that the social conditions of participants would be the consequence, rather than a cause, of depressed mood. E.g. a depressed mood might have hindered people to have contacts with other people, increasing the risk of becoming 'marginalised'. Longitudinal data will be required to demonstrate whether the prevalence of depressed mood will decrease if social conditions for the second generation improve.

A second limitation is the use of self-reported data on depressed mood, based on the PHQ-9 among different ethnic groups. A recent analysis on the measurement invariability of the PHQ-9 showed that the PHQ-9 items were measuring the same underlying construct in the different ethnic groups included in our study [27]. That suggests that PHQ-9 is a good measure of depressive symptoms in those groups. The PHQ-9 outcome is not equivalent, though, to a diagnosis of clinical depression, implying that our results are not automatically generalisable to an outcome of clinical depression.

Third, we were not able to draw firm conclusions about whether the prevalence of depressed mood in immigrants' offspring was lower than or similar to that in people of the first generation. The indirect comparison, whereby the host population was used as the reference group to both the immigrants and the immigrants' offspring, led us to conclude that the risk in the offspring was slightly less elevated than that in the immigrants. However, in view of

the minimal overlap in the age distributions of the two generations, that conclusion may be biased by an age effect. A direct comparison of prevalence in both generations might have been more reliable, but it yielded wide confidence intervals, probably related to the age distribution differences. To enable conclusions on changes in risk across generations, future research should include both first and second generations and ensure, if possible, substantial overlap in age range.

## Interpretations and implications

In all four minority ethnic groups we studied, the prevalence of depressed mood in immigrants' offspring was higher than that in the host population. In contrast with the 'immigrant health paradox', a similar pattern was observed for immigrants. This result also contrasts with several previous studies that have found comparable rates of depression in immigrant offspring and host populations. Studies based on the European Social Survey, for instance, found no elevated risk for immigrant offspring in Europe [1,8]. Those studies, however, combined data from ethnic minority populations into one category, thereby possibly masking differences between ethnic groups. A study in Australia, as a case in point, has indicated a higher prevalence of mental health problems in the offspring of immigrants of European descent, but not of other ethnic groups [14]. This highlights the importance of distinguishing by ethnic background in future studies.

The size of risk differed between ethnic groups in our own study, in fact, with offspring of Turkish descent showing the highest risk and those of African Surinamese descent the lowest. Since we found similar within-group risk patterns for immigrants and offspring, that could point to risk factors inherent in particular ethnic groups, such as culture or genetic profiles. The higher risk of depression in offspring of parents with depression is well documented [36], and it could be linked to genetic factors or to family interaction (communication, experience of affect). Such factors could partially underlie similarities in pattern across generations in a particular ethnic group. In addition, the differences in size of the risk between ethnic groups might reflect the prevalence of depressed mood in the country of origin [4]. The prevalence of this health problem has been shown to differ between regions of the world, with, e.g., a relatively low prevalence in Sub Saharan countries where the African Suriname originally come from [37].

This is not to say, of course, that a risk of depression in a particular ethnic group is determined entirely by attributional factors–unique and rather stable attributes of that group such as genetics and culture [22]. Quite the contrary, our study has revealed that the higher risk of depression for immigrants and their offspring in ethnic minority groups is strongly associated with the living conditions to which the groups are exposed in the host country, so-called postmigration factors. This is in line with previous studies that show the salience of such social conditions for immigrants themselves. To the best of our knowledge, we are the first to study the impact of multiple types of social conditions simultaneously on the burden of depressed mood in immigrants' *offspring*. Similarly to the results in previous studies of immigrants, we found that the higher risk in the offspring was accounted for by social conditions to which they were subject, in particular an adverse socioeconomic position and the experience of discrimination. Should that be confirmed in future studies using longitudinal data, it will be an indication that improving the living conditions of immigrants' offspring–and in particular their socioeconomic position and their exposure to discrimination–could be a promising strategy to reduce depression rates in the offspring generations.

Whereas in our study the greater risk of depression in immigrants' offspring could be fully explained by adverse socioeconomic conditions and discrimination, a higher risk partially

persisted in immigrants after those social conditions had been taken into account. A small part of their remaining higher risk was explained by sociocultural factors, whereas the analysis for immigrants' offspring left no room for additional contributions by sociocultural factors. Important for this difference seems the higher percentage of first-generation respondents who were marginalised or separated–subgroups that showed higher depression risk than immigrants classified as integrated. Apart from the minor difference in risk levels, though, it is important to note that the distribution of immigrants' offspring across acculturation categories was largely similar to that of the immigrants themselves. In addition, also the risk of depression associated with specific acculturation categories was largely similar across generations, implying that also for the immigrants' offspring, the process of acculturation might be as stressful as it is for first generation immigrant.

Also worth highlighting, and in line with the results of a meta-analysis by Yoon and colleagues [12], is that our respondents who were classified as integrated–that is, showing an orientation towards the cultures of both the host country and the country of origin–had the lowest risks of depressed mood. They thus showed a lower risk than those classified as assimilated (integrated into Dutch society with a low orientation to the country of origin). This provides indications that it is beneficial for people from ethnic minority populations to stay attached to the original country's culture and people. Whereas this has been put forward as an explanation for the relative good mental health of immigrants who fit the 'healthy immigrant paradox' [14], our results provide indications to suggest that this beneficial effect also holds for immigrant's offspring.

In our statistical analysis, we dealt with social conditions as three separate categories–socioeconomic position, discrimination and sociocultural situation. In reality, of course, such conditions are interlinked, and they could have an interactive impact on mental health. People without paid jobs, for example, may have less contact with people from the host country, with a consequent influence on their acculturation strategy. Similarly, also experiences of discrimination might impact on the process of acculturation [20]. Socioeconomic position can also impact mood through cultural factors such as low cultural entitlement–a feeling of not 'being a relevant and legitimate citizen who matters in society' [5]. Moreover, multiple categories of social conditions may share the same underlying mechanisms, such as institutional racism [38]. For example, institutional racism may reduce the likelihood that people from ethnic minority populations will get a job, and thereby weaken their sense of affiliation with the host society's culture. Unravelling the complex ways in which such conditions interact, and probably also have synergic effects, is crucial if the aim is to counter the high risk of depressed mood in ethnic minority populations through improving their social conditions. The results of our study suggest that the way in which these conditions interact might be different for immigrants and immigrants' offspring, given, e.g. the fact that, in our multivariable model, sociocultural conditions did not contribute to the increased risk of immigrants' offspring whereas it did for immigrants. As we found similar associations in the univariate model, this might indicate that sociocultural conditions impact on mental health in the immigrant's offspring through socioeconomic conditions. The differential impact of discrimination on the process of acculturation between generations is another example of how mechanisms might differ for immigrants and immigrants' offspring [20].

## Conclusion

The prevalence of depressed mood in the offspring of immigrants in the Netherlands was found to be substantially higher than that in the Dutch host population. Our results provide indications that their depression risk will decline to the Dutch level as the various populations

grow closer in terms of socioeconomic position and as immigrant offspring cease to experience discrimination.

## Supporting information

**S1 Table. Associations between indicators for social conditions and depressed mood in first-generation immigrants, by ethnic group.**
(DOCX)

**S2 Table. Contribution of social indicators to the probability of depressed mood in immigrants' offspring and immigrants relative to those of Dutch origin.**
(DOCX)

## Acknowledgments

The HELIUS study is conducted by the Academic Medical Center Amsterdam and the Public Health Service of Amsterdam. We are most grateful to the participants of the HELIUS study and the management team, research nurses, interviewers, research assistants and other staff who have taken part in gathering the data of this study. The authors also thank Michael Dallas for editing the manuscript.

## Author Contributions

**Conceptualization:** Karien Stronks, Aydın Şekercan, Marieke Snijder, Anja Lok, Arnoud P. Verhoeff, Anton E. Kunst.

**Data curation:** Marieke Snijder.

**Formal analysis:** Karien Stronks, Marieke Snijder, Henrike Galenkamp.

**Investigation:** Karien Stronks, Aydın Şekercan.

**Methodology:** Karien Stronks, Marieke Snijder, Anja Lok, Arnoud P. Verhoeff, Anton E. Kunst, Henrike Galenkamp.

**Supervision:** Anton E. Kunst, Henrike Galenkamp.

**Writing – original draft:** Karien Stronks, Marieke Snijder.

**Writing – review & editing:** Aydın Şekercan, Marieke Snijder, Anja Lok, Arnoud P. Verhoeff, Anton E. Kunst, Henrike Galenkamp.

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
