## [Decision Letter · Decision Letter 0]

19 Mar 2020

PONE-D-19-34059

Higher prevalence of depressed mood in immigrants’ offspring reflects their social conditions in the host country: The HELIUS study

PLOS ONE

Dear Dr. Stronks,

Thank you for submitting your manuscript to PLOS ONE. After careful consideration, we feel that it has merit but does not fully meet PLOS ONE’s publication criteria as it currently stands. Therefore, we invite you to submit a revised version of the manuscript that addresses the points raised during the review process.

We would appreciate receiving your revised manuscript by May 03 2020 11:59PM. To enhance the reproducibility of your results, we recommend that if applicable you deposit your laboratory protocols in protocols.io, where a protocol can be assigned its own identifier (DOI) such that it can be cited independently in the future. For instructions see: http://journals.plos.org/plosone/s/submission-guidelines#loc-laboratory-protocols

We look forward to receiving your revised manuscript.

Kind regards,

Ali Montazeri

Academic Editor

PLOS ONE

Journal Requirements:

"The HELIUS study is conducted by the Academic Medical Center Amsterdam and the Public Health Service of Amsterdam. Both organisations provided core support for HELIUS. The HELIUS study is also funded by the Dutch Heart Foundation, the Netherlands Organization for Health Research and Development (ZonMw), the European Union (FP-7), and the European Fund for the Integration of non-EU immigrants (EIF)."

Reviewers' comments:

Reviewer's Responses to Questions

**Comments to the Author**

1. Is the manuscript technically sound, and do the data support the conclusions?

Reviewer #1: Partly

2. Has the statistical analysis been performed appropriately and rigorously? 

Reviewer #1: I Don't Know

3. Have the authors made all data underlying the findings in their manuscript fully available?

Reviewer #1: No

4. Is the manuscript presented in an intelligible fashion and written in standard English?

Reviewer #1: Yes

5. Review Comments to the Author

Reviewer #1: The authors use a large-scale cohort study to compare the social risk factors for depressed mood between first and second generation immigrants. The data they use is impressive, as is their ability to disaggregate into four different country of origin groups. Overall, I would like to see more engagement with the immigrant health literature and a more detailed explanation for their statistical models.

The authors emphasize their examination of social risk factors for depressive symptoms among immigrant offspring as their contribution to the literature, but I would not agree that they are the first to study this topic. There are many studies on acculturation and health among second generation immigrants and the loss of the “immigrant paradox” (which primarily tests the salience of SES on health) in the second generation. Perhaps they are the first to study this specific outcome, but they can still engage more with this established field in the Introduction and Discussion. Some example papers:

• Cervantes, R. C., Padilla, A. M., Napper, L. E., & Goldbach, J. T. (2013). Acculturation-Related Stress and Mental Health Outcomes Among Three Generations of Hispanic Adolescents. Hispanic Journal of Behavioral Sciences, 35(4), 451–468. https://doi.org/10.1177/0739986313500924

• Renee Luthra, Alita Nandi & Michaela Benzeval (2018) Unravelling the ‘immigrant health paradox’: ethnic maintenance, discrimination, and health behaviours of the foreign born and their children in England, Journal of Ethnic and Migration Studies, DOI: 10.1080/1369183X.2018.1539287

• Molly A. Martin, Jennifer L. Van Hook, Susana Quiros. (2015) Is socioeconomic incorporation associated with a healthier diet? Dietary patterns among Mexican-origin children in the United States. Social Science & Medicine, Volume 147. Pages 20-29.

A related comment is that I would like to see more engagement with the immigrant health literature overall. The authors don’t discuss the immigrant health paradox or health selection as possible interpretation to their findings. They also don’t provide any discussion, either in the Introduction or Discussion, about why they might expect to see subgroup differences (health selection would fit in well here). The country of origin differences are underdeveloped.

The results are hard to follow. I’m not sure why the authors chose to provide their results solely in AMEs and not in odds ratios, which are more easily understood by readers. Tables 3 and A are particularly hard to digest. It’s not clear which logistic regression model these AMEs are derived from. They say in the methods section that they added the social conditions one by one, which makes me think these variables were included sequentially. Can the authors at least include the coefficients of the logistics regression? Perhaps they should consider graphing the results of Table 3; trends would be much easier to detect.

I would like more information about how to interpret the final column in Table 2. They need to provide more detail in how they calculated this differences and how to interpret it. In general, I would like to see a lot more detail about their analyses. The reason why my response for #2 is "I don't know' is because I don't feel I am able to evaluate with the information they have provided in this version. Were the AMEs calculated at observed values, means, etc? Again, I would like to see the coefficients of the model, even as an Appendix.

I'm concerned with the missing data for occupational level. Given the extremely high level of missing for some subgroups, I’m not sure how to interpret what this group means. I would suggest limiting their socioecomic measures to variables with less missing information: education level and employment status.

The authors don’t make a strong case for they why chose the measures for each of their social conditions. Sociocultural factors, for example, have three different measures. There are certainly some endogeneity problems with the cultural orientation measures; someone who is “marginalized” probably has underlying mental health issues to begin with and does not indicate any causal effect of acculturation. I would like to see more more justification for these measures.

6. PLOS authors have the option to publish the peer review history of their article (what does this mean?). If published, this will include your full peer review and any attached files.

Reviewer #1: No

---

## [Author Response · Author response to Decision Letter 0]

24 Apr 2020

Reply to Editor’s comments 

>> We have now included a data availability statement at p. 23 of the manuscript.

"The HELIUS study is conducted by the Academic Medical Center, and the Public Health Service of Amsterdam. Both organisations provided core support for HELIUS. The HELIUS study is also funded by the Dutch Heart Foundation, the Netherlands Organization for Health Research and Development (ZonMw), the European Union (FP-7), and the European Fund for the Integration of non-EU immigrants (EIF)."

>> We would like to include the following statement in the Funding Statement:

The HELIUS study is funded by the University Medical Centers Amsterdam, location AMC, and the Public Health Service of Amsterdam, as well as by the Dutch Heart Foundation, the Netherlands Organization for Health Research and Development (ZonMw), the European Union (FP-7), and the European Fund for the Integration of non-EU immigrants (EIF).

>> As we believe this paragraph on the stratified analyses by gender is not the core of the paper, we have now removed this limitation. 

>> We have made the changes you asked for.

Reply to Reviewers' comments

Reviewer #1: The authors use a large-scale cohort study to compare the social risk factors for depressed mood between first and second generation immigrants. The data they use is impressive, as is their ability to disaggregate into four different country of origin groups. Overall, I would like to see more engagement with the immigrant health literature and a more detailed explanation for their statistical models.

>> Thank you for your kind words, as well as for your comments and suggestions, which were very helpful in improving the paper.

The authors emphasize their examination of social risk factors for depressive symptoms among immigrant offspring as their contribution to the literature, but I would not agree that they are the first to study this topic. There are many studies on acculturation and health among second generation immigrants and the loss of the “immigrant paradox” (which primarily tests the salience of SES on health) in the second generation. Perhaps they are the first to study this specific outcome, but they can still engage more with this established field in the Introduction and Discussion. Some example papers:

• Cervantes, R. C., Padilla, A. M., Napper, L. E., & Goldbach, J. T. (2013). Acculturation-Related Stress and Mental Health Outcomes Among Three Generations of Hispanic Adolescents. Hispanic Journal of Behavioral Sciences, 35(4), 451–468. https://doi.org/10.1177/0739986313500924

• Renee Luthra, Alita Nandi & Michaela Benzeval (2018) Unravelling the ‘immigrant health paradox’: ethnic maintenance, discrimination, and health behaviours of the foreign born and their children in England, Journal of Ethnic and Migration Studies, DOI: 10.1080/1369183X.2018.1539287

• Molly A. Martin, Jennifer L. Van Hook, Susana Quiros. (2015) Is socioeconomic incorporation associated with a healthier diet? Dietary patterns among Mexican-origin children in the United States. Social Science & Medicine, Volume 147. Pages 20-29.

A related comment is that I would like to see more engagement with the immigrant health literature overall. The authors don’t discuss the immigrant health paradox or health selection as possible interpretation to their findings. They also don’t provide any discussion, either in the Introduction or Discussion, about why they might expect to see subgroup differences (health selection would fit in well here). The country of origin differences are underdeveloped.

>> Thank you for your suggestions. Firstly, as indicated in the introduction, we do not claim that we are the first to study the prevalence of mental health problems in immigrants’ offspring. We referred to previous studies, but were not aware of the abovementioned study by Cervantes et al. Thank you for bringing that to our attention, we have now included it in the list of references. 

Secondly, we are indeed aware of the literature on the ‘healthy immigrant paradox’. However, as this paradox stipulates that immigrants experience better health, we believe it does not apply to mental health outcome in Europe, in contrast with the outcome of studies in North America. We have now mentioned this contrast in the introduction as well as the discussion section. 

Thirdly, we agree that we could engage more with the current literature, even though previous studies do not exactly correspond with ours. We have now incorporated references to current literature on multiple aspects of the topic of our paper (including healthy migrant paradox, acculturation stress, post migration factors etc), both in the introduction and discussion section. 

The results are hard to follow. I’m not sure why the authors chose to provide their results solely in AMEs and not in odds ratios, which are more easily understood by readers. 

>> We used AME’s as outcome of the logistic regression models, given that our aim was to compare the contribution of social determinants across groups, in particular first and second generation, but also ethnic groups. When making comparisons across groups, ORs can be misleading as an indicator of the size of the contribution of each predictor as a large change in odds does not always correspond with a large change in predicted probability. In contrast, AMEs do allow for a fair comparison of effect sizes across groups [Mood, C. (2010). Logistic regression: Why we cannot do what we think we can do, and what we can do about it. European sociological review, 26(1), 67-82.]. As readers will indeed probably more used to the use of ORs, we have now explained in the methods section why ORs can be misleading, and included extra information as to how to interpret this measure. 

Tables 3 and A are particularly hard to digest. It’s not clear which logistic regression model these AMEs are derived from. 

>> We apologize for the unclarity. We have now added a more precise description of the models in the headings of both tables as well in the accompanying text in the result section. In addition, we have added a footnote as to how to interpret the AME. Finally, for each indicator, we have now also included the reference category. 

They say in the methods section that they added the social conditions one by one, which makes me think these variables were included sequentially. 

>> We apologize for the confusion. Indeed, we added the social conditions one by one. However, as indicated in Figure 1 and Table B (appendix), we also added these as a set (i.e. socioeconomic, perceived discrimination and sociocultural factors respectively). We have now further clarified this in the methods section. In addition, we have clarified the presentation of these models in the results section. 

Can the authors at least include the coefficients of the logistics regression? 

>> For our reply to this suggestion, we refer to our answer below.

Perhaps they should consider graphing the results of Table 3; trends would be much easier to detect.

>> Thank you for this suggestion. We have considered graphing the results given in Table 3, but given our wish to show the outcome for seven indicators in 4 ethnic groups, such a graph would become rather large. In addition, it would be more difficult to get a clear overview of the pattern of the confidence intervals for all indicators. We therefore propose to keep the table. 

I would like more information about how to interpret the final column in Table 2. They need to provide more detail in how they calculated this differences and how to interpret it. 

>> We again apologize for the confusion. We have now added more information in the heading and in Table 2 itself, to further clarify the comparisons we make in the three columns. To further simplify Table 2, we have now moved the unadjusted percentages of depressed mood in each of the ethnic groups from Table 2 to Table 1. 

In general, I would like to see a lot more detail about their analyses. The reason why my response for #2 is "I don't know' is because I don't feel I am able to evaluate with the information they have provided in this version. Were the AMEs calculated at observed values, means, etc? Again, I would like to see the coefficients of the model, even as an Appendix.

>> We hope the further clarification we provided on the measurement (data & methods section) and interpretation (results section) of AMEs now gives enough information to fully understand the results. We are hesitant to present the coefficients of the model, exactly for the reason that we gave to prefer AME instead: coefficients can be misleading as an indicator of effect size as a large change in coefficient does not always correspond with a large change in predicted probability. 

I'm concerned with the missing data for occupational level. Given the extremely high level of missing for some subgroups, I’m not sure how to interpret what this group means. I would suggest limiting their socioecomic measures to variables with less missing information: education level and employment status.

>> The missing data for occupational level largely reflects the situation of people (and/or their partner) who have never had a paid job. When we studied the contribution of each socio-economic variable to the model (S2 Table B), the contribution of each of these variables appeared to be similar. For that reason, it would not make a difference in result when we followed the suggestion of the reviewer to limit the socioeconomic variables to income and education only. As the term ‘missing’ wrongly suggests a data problem, we have now changed this heading into ‘not applicable’. 

The authors don’t make a strong case for they why chose the measures for each of their social conditions. Sociocultural factors, for example, have three different measures. There are certainly some endogeneity problems with the cultural orientation measures; someone who is “marginalized” probably has underlying mental health issues to begin with and does not indicate any causal effect of acculturation. I would like to see more more justification for these measures.

>> We have now added a justification for the (multiple) measures we chose for socioeconomic position and sociocultural conditions. In addition, we agree that the association between at least some of these measures and depressed mood might also reflect a reverse effect. We had already alluded to this type of bias in the limitations section, and have now added the example of marginalisation to illustrate this. Unfortunately, this bias reflects the cross-sectional design of the study, and cannot be solved by choosing other measures for sociocultural orientation.

---

## [Editor Report · Decision Letter 1]

18 May 2020

Higher prevalence of depressed mood in immigrants’ offspring reflects their social conditions in the host country: The HELIUS study

PONE-D-19-34059R1

Dear Dr. Stronks,

We are pleased to inform you that your manuscript has been judged scientifically suitable for publication and will be formally accepted for publication once it complies with all outstanding technical requirements.

With kind regards,

Ali Montazeri

Academic Editor

PLOS ONE
---

## [Editor Report · Acceptance letter]

26 May 2020

PONE-D-19-34059R1 

Higher prevalence of depressed mood in immigrants’ offspring reflects their social conditions in the host country: The HELIUS study 

Dear Dr. Stronks:

I am pleased to inform you that your manuscript has been deemed suitable for publication in PLOS ONE. Congratulations! Your manuscript is now with our production department. 

With kind regards,

on behalf of

Professor Ali Montazeri 

Academic Editor

PLOS ONE